TECHNICAL RELEASE

# DAPT: A package enabling distributed automated parameter testing

Ben Duggan[1], John Metzcar[1] and Paul Macklin[1,*]

1  Indiana University Luddy School of Informatics, Computing and Engineering, 107 S Indiana Ave, Bloomington, IN 47405, USA

## ABSTRACT

Modern agent-based models (ABM) and other simulation models require evaluation and testing of many different parameters. Managing that testing for large scale parameter sweeps (grid searches), as well as storing simulation data, requires multiple, potentially customizable steps that may vary across simulations. Furthermore, parameter testing, processing, and analysis are slowed if simulation and processing jobs cannot be shared across teammates or computational resources. While high-performance computing (HPC) has become increasingly available, models can often be tested faster with the use of multiple computers and HPC resources. To address these issues, we created the Distributed Automated Parameter Testing (DAPT) Python package. By hosting parameters in an online (and often free) "database", multiple individuals can run parameter sets simultaneously in a distributed fashion, enabling *ad hoc* crowdsourcing of computational power. Combining this with a flexible, scriptable tool set, teams can evaluate models and assess their underlying hypotheses quickly. Here, we describe DAPT and provide an example demonstrating its use.

**Subjects**  Software and Workflows, Software Engineering

**Submitted:**    01 March 2021

\* Corresponding author. E-mail: macklinp@iu.edu

Preprint submitted at https://doi.org/ 10.20944/preprints202103.0116.v2

## STATEMENT OF NEED

### Introduction

Evaluating a new computational model requires testing many parameter sets and validating the results [1, 2], collectively called model exploration (ME) [3]. For complex models with many parameters to explore, computational time can be high and managing the testing pipeline, processing the results, and storing the data can quickly become cumbersome. To mitigate this, tools to facilitate ME on high-performance computing (HPC) resources such as Extreme-scale Model Exploration with Swift (EMEWS) [4] and Open MOdeL Experiment (OpenMOLE) [5], have been developed. EMEWS and OpenMOLE distribute large scale ME jobs to HPC systems. Additionally, they can adaptively explore parameter spaces to achieve some predefined simulation outputs or goals.

However, there are several complications with HPC and by extension these ME software packages. The "headless" (non-graphical) nature of HPC means that people unfamiliar with command-line terminals may struggle to utilize the resources. This can be a particular challenge that slows onboarding for multidisciplinary team members, or people less familiar with servers. Sharing data produced from a simulation created on an HPC can also be challenging. After prototyping simulation models and analysis workflows on desktop workstations, it can be time consuming to adapt them to HPC resources, particularly for

applications that require a graphical user interface (GUI) or software not supported on the HPC platform. Finally, not all teams may have low-cost access to HPC and cloud compute resources.

As a result, many teams come to rely upon a single member to run the model exploration, either on a personal computer or HPC. Since compute and processing times may already take up a considerable portion of project time, concentrating this work on one team member or one compute system compounds this already existing problem. One way to combat this is by splitting up the parameter sets among the team (or a broader community), having each team or community member run them on their computer or HPC resource, and then uploading the results to a shared storage solution. This distributed computational approach has been automated and shown to be effective on large scale projects such as Folding@Home (F@H) [6]. F@H uses a community of people and organizations who volunteer their computational resources to simulate protein folding. However, F@H is not ideal for small groups because the code is closed-source, requiring the team to develop the software anew, and requires the use of servers to assign jobs. Moreover, F@H is tailored to one specific scientific problem; it was not used to facilitate independent third-party scientific workflows.

There are many ways to leverage distributed computing for model testing. For example, F@H uses a client-server architecture. With this approach, clients (computers operated by community volunteers) get simulation parameters from a job distribution server. This server also maintains a database of simulation parameters and job statuses. An alternative is a database-centric design. In this approach, each client interacts directly with the database to gather parameters and update the job status. This second method removes the need for a centralized server, making setup and maintenance much simpler, as only the database needs to be managed. Furthermore, depending on the database requirements, there are many freely available cloud platforms which can be used to store parameters. For example, Google Sheets can be used as an online "database" that stores tests in each row and parameters in each column.

To address the issues discussed above, we created DAPT (Distributed Automated Parameter Tester). In particular, we aimed to (1) make ME more broadly accessible to small teams with diverse programming backgrounds through a simple Python library, (2) allow small teams to pool their individual computational resources to perform concurrent, distributed ME using a database-centric architecture, and (3) provide easy integration of "off-the-shelf" cloud resources and storage services for simple inclusion in ME pipelines. By adhering to these design principles, once the workflow is created, new teammates or even those simply with idle computing resource can contribute to a team's parameter studies through straightforward code sharing. Thus, DAPT allows for *ad hoc* crowdsourcing of computational power to create a small-scale, F@H-like testing environment.

Computational models require large amounts of parameter testing and simulations to explore and validate a model. To our knowledge, there are no software packages that allow pipelines to easily connect with application programming interfaces (APIs) and enable serverless *ad hoc* crowdsourcing of computing power. We created DAPT to allow easy integration of low-cost (or free) cloud services (e.g. Google Sheets and Box) into ME pipelines and enable all members of a team to pool their computing resources to run simulations, rather than just one person.



**Table 1.** A description of the main components of DAPT along with an example showing how to use the component.

| Class/Module | Description | Required | Example accessing |
|---|---|---|---|
| db | The `db` package contains many different databases. These databases store the parameters and job metadata and have methods to get and update the values. | Yes | `db = dapt.db.Delimited_file('database.csv')` Will create a 'Delimited_file' database from the 'database.csv' file. |
| Param | The `Param` class is the main class that users interact with. It is responsible for getting and updating tests. | Yes | `param = dapt.Param(db)` Creates a parameter object using the database `db`. The next parameters can then be obtained by calling `param.next_parameters()`. |
| storage | The `storage` package contains several modules which make uploading and downloading files or folders easy. There are several services where data can be obtained from within the `storage` package. | No | `box = dapt.storage.Box(config=config)` This creates a `Box` class object which, after being initialized by calling `box.connect()`, allows for the user to interact with their files on Box. |
| Config | The `Config` class uses a JSON file to store testing settings, API credentials, and user custom parameters. It is not required, but using the `Config` class makes DAPT easier to use. | Recommended | `config = dapt.Config('config.json')` This will create a `Config` object from the `config.json` file. If the value `user-name` was in the JSON file then it could be retrieved by calling `config.get_value('user-name')` |
| tools | A collection of tools that make DAPT easier to use, especially with PhysiCell. | No | `dapt.tools.sample_db()` creates a `Delimited_file` database. |

## IMPLEMENTATION

DAPT is written and tested for Python versions 3.6 through 3.9.1. It was written modularly, allowing users to call individual DAPT components and create their own custom pipeline. It is imported by adding `import dapt` at the top of the user's script. The main components of DAPT are shown in Table 1. The `db` (database) package contains modules capable of storing parameters to be tested and simulation and job metadata. The `Param` class interacts with a database class instance to get parameters to test, update test metadata, and mark when tests finish. Figure 1 shows how DAPT is used to test a model and how multiple team members can contribute simultaneously.

Parameters to test (and accompanying meta-data) are stored in a `Database` object. There are many databases available in the `db` package, which all inherit the parent `Database` (dapt.db.Database) class, ensuring compatibility with core methods. Database objects follow a non-relational scheme where parameters are stored in a table. The table is similar to a spreadsheet where rows hold an individual parameter set and job information and a column holds the same attribute or parameter with potentially different values. The head of the table holds the attribute names or the names of each parameter. An example database can be seen in Table 2. Databases have a method named `get_table()` which retrieves the entire set of parameters. The method `get_attributes()` returns the attributes of the database. Lastly, the methods `update_row()` and `update_cell()` update an entire row or specific entry, respectively. The current database options are delimited files (such as a comma separated values [CSV] file) or free online spreadsheet applications. Delimited files work well for a single user, whereas online spreadsheets—which are notable for their user-friendly interface—are required to use DAPT in its distributed mode.

Each parameter set is encoded into a job, corresponding to one row in the database. Each job must have a unique `id` associated with it and a `status` attribute. The `id` attribute is used as a unique job identifier, and the status stores which job task is currently being completed. The `status` field is initially empty and is updated as the job proceeds. The value "successful" indicates a job is complete, "failed" means the job finished unsuccessfully, and any other value shows which task of the user-defined pipeline the job is currently completing. Other attributes can be included in the table to add additional information. For example, the `start-time` attribute stores the time that a job was started. A complete list of Database

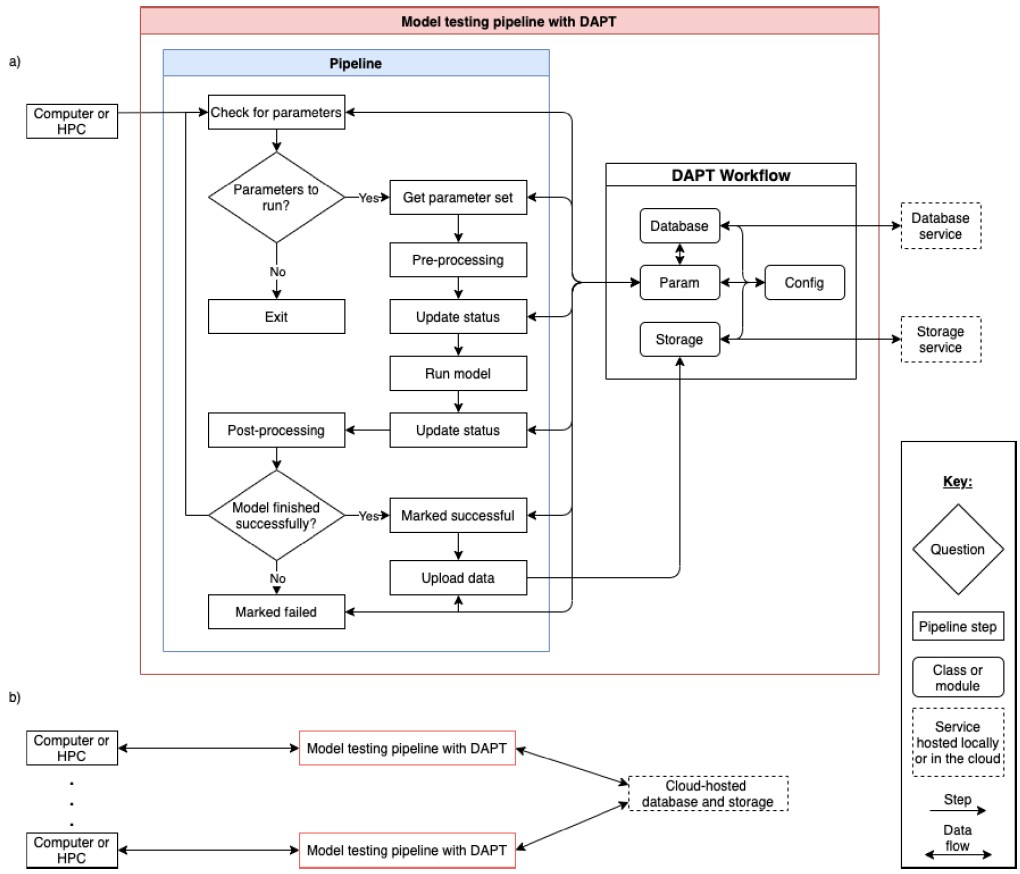

**Figure 1.** Overview of typical model exploration pipeline using DAPT. (a) Shows how one resource can be used to run the testing pipeline. The database and storage module can be local or cloud-hosted. (b) Multiple team members or resources can be used to run the pipeline in a distributed manner. To test parameter sets collaboratively, the database and storage option must be hosted online.

attributes is found in the documentation [7]. To test stochastic models that require multiple runs at the same location in parameter space, parameter sets can be re-run in as many new, unique jobs as needed.

The class that brings all the components together is the `Param` class, short for Parameter. The `Param` class interacts with the Database instance to manage the compute jobs. The next parameters to be tested are retrieved using the `next_parameters()` method. This method returns the parameters from the next entry in the database with an empty status attribute. Other constraints can be placed on this method, such as the required computational power to run a parameter set. This method also marks the `status` as "in progress" and populates any related fields present (e.g. `start-time`). This ensures that the job is not run twice. The other methods of the `Param` class require the current `id` of the job be given. This means that attributes of jobs other than the current job can be modified. For example, if your current job `id` is "test1", there are no restrictions preventing you from manipulating the attributes of "test2". As a consequence, only people you trust should participate in the crowdsourced computing. The status of a job can be set using the `update_status()` method. The

`successful()` and `failed()` methods are used to mark that a job was completed successfully or with a problem, respectively.

DAPT also makes it easy to interact with cloud storage providers through the Storage (`storage`) package. These modules support uploading, downloading, deleting, and renaming files. While not required for core functionality, these modules allow data to be easily uploaded to a shared location, or downloaded for a job or further processing. This automated sharing facilitates discussion of simulation results and may enhance real-time collaboration. Classes in the `storage` package must inherit the `Storage` (dapt.storage.Storage) class which outlines the required methods. API credentials must be created by each user. There are guides posted online for each `storage` API offered.

The last component discussed is the Config class. This class takes in a JavaScript Object Notation (JSON) file and creates a dictionary or array from the contents of the file. The Config class can be used by DAPT to store information about APIs (e.g. the Google Sheet ID), how a job should be run (e.g. skip jobs meant for HPCs), how a set of jobs should be run (e.g. the number of jobs to run before quitting) and data that gets updated and persists between running (e.g. the last job id and API tokens). Instances of this class are used by all DAPT classes, can also be used to store information for jobs, and makes initialization of classes easy. This class also contains methods to update the JSON file so the changes persist to other simulation runs.

## Example

There are many basic examples provided with DAPT that demonstrate how each module functions. They can be found in the examples folder of the GitHub repository [8]. To provide a real-world example of how DAPT can be used, we will use PhysiCell [9] version 1.7.0, an open-source, agent-based multicellular simulation framework. No knowledge of PhysiCell is necessary to understand the example. In this example, we use the "biorobots" sample project (included with every PhysiCell download) where "worker" cells drag biological cargo towards "director" cells. More information about the model along with a GUI to explore the parameters is available through the PhysiCell biorobots simulation tool on nanoHUB [10]. The code for this example can be found in the `paper_example.py` file in the DAPT example for this paper [11].

When creating a PhysiCell model, diffusion and cell parameters are defined in the C++ code and loaded from an Extensible Markup Language (XML) file. A sample of a PhysiCell settings file is shown in Listing 1. There are several parameters, but the parameters of interest for our example are `<attached_worker_migration_bias>` and `<unattached_worker_migration_bias>`, located within the `<user_parameters>` tag. These parameters range from zero to one. As the bias approaches zero, the cell migration path approaches a random walk, while cell migration paths become more directed and deterministic as the bias approaches one.

In this example, the XML tags will be represented as a path from the root of the file. For instance, the `<attached_worker_migration_bias>` tag is represented as `/user_parameters/attached_worker_migration_bias`. Using the `dapt.tools.create_XML()` function, a dictionary containing these paths can be used to update the XML settings file. The keys in the dictionary are paths with parameter values as the values. This method is beneficial, as the code necessary to update the settings is not hard-coded. Another attribute could then be added to the database without changing the testing script.



```
1   <PhysiCell_settings version="devel-version">
2           <domain> ... </domain>
3           <overall> ... </overall>
4           <microenvironment_setup> ... </microenvironment_setup>
5           <user_parameters>
6                   <attached_worker_migration_bias type="double">1.0</attached_worker_migration_bias>
7                   <unattached_worker_migration_bias type="double">0.5</unattached_worker_migration_bias>
8           </user_parameters>
9   </PhysiCell_settings>
```

**Listing 1.** The skeleton of a PhysiCell settings file. The "attached_worker_migration_bias" is a custom variable which changes the migration bias of workers attached to cargo.

**Table 2.** Parameters used for PhysiCell example. The head of the table holds the attributes used. Each row is a different job to be run, each with its own id, metadata, and parameter set. This table is stored as a CSV named "parameters.csv" which DAPT uses as the database.

| id | status | start-time | end-time | comment | attached_worker_bias* | unattached_worker_bias[†] |
|---|---|---|---|---|---|---|
| default | | | | | 1.0 | 0.5 |
| attached | | | | | 0.1 | 1.0 |
| unattached | | | | | 1.0 | 0.1 |

*Full path "/user_parameters/attached_worker_migration_bias". [†]Full path "/user_parameters/unattached_worker_migration_bias".

For this example, three jobs will be run as shown in Table 2. As explained earlier, the `id` and `status` attributes are required. The `start-time`, `end-time`, and `comment` attributes are optional, but they provide additional information. These parameters are saved in a comma separated values (CSV) file named `parameters.csv`. This file is updated as the jobs run, showing the progress that has been made.

The code for this example is shown in Listing 2. The three DAPT modules that are used are `Config`, `Delimited_file`, and `Param`. The configuration for this example is stored in `config.json` and has two options: `last-test` and `num-of-runs`. The first option is used to store the current job `id`, which is needed for DAPT to resume a job if the program crashes or is stopped. The second option allows the number of jobs to run to be specified which is all jobs in this case. The full contents of the config file should be: `"last-test":null,` `"num-of-runs":-1`, saved as "config.json".

The folder structure of this project has the Python script (Listing 2), config.json, and parameters.csv inside the PhysiCell directory. The first two lines of code import the required modules. The `os` module is used for interacting with the file system and the `platform` module is used to detect which operating system is being used. `dapt` imports all of the DAPT modules needed. Lines four through six instantiate the three DAPT modules needed. The config file is passed to the `Param` class, enabling the settings to be used.

The next line gets the parameter set using the `next_parameters()` method. If there are no more parameters to run and thus no more jobs in the database, then `None` is returned. Lines 10 through 21 contains the main pipeline. This starts with a while loop that checks to see if there are more parameters to run. The next line uses the `create_XML()` method to load the parameters into the settings file. The status of the parameter set is then updated on line 13. Lines 15 through 18 check if the operating system is Windows or Unix-based, as different operating systems run executable files differently. The last two lines mark this job as complete and gets the next set of parameters. The outputs from the PhysiCell simulation are shown in Figure 2 and the contents of the CSV file after finishing the all jobs are shown in Table 3.



```
1    import os, platform
2    import dapt
3
4    config = dapt.Config(path='config.json')
5    db = dapt.db.Delimited_file('parameters.csv', delimiter=',')
6    params = dapt.Param(db, config=config)
7
8    p = params.next_parameters()
9
10   while p is not None:
11       dapt.tools.create_XML(p, default_settings="PhysiCell_settings_default.xml", save_settings="PhysiCell_settings.xml")
12
13       params.update_status(p["id"], 'running simulation')
14
15       if platform.system() == 'Windows':
16           os.system("biorobots.exe")
17       else:
18           os.system("./biorobots")
19
20       params.successful(p["id"])
21       p = params.next_parameters()
```

**Listing 2.** An example showing how DAPT can be used to perform parameter testing on an agent-based models such as those written in PhysiCell.

**Table 3.** The PhysiCell example CSV file from Table 2 updated after running DAPT.

| id | status | start-time | end-time | comment | attached_worker_bias* | unattached_worker_bias† |
|----|--------|------------|----------|---------|----------------------|------------------------|
| default | successful | 2021-02-28 19:36:30 | 2021-02-28 19:38:20 | | 1.0 | 0.5 |
| attached | successful | 2021-02-28 19:45:09 | 2021-02-28 19:45:09 | | 0.1 | 1.0 |
| unattached | successful | 2021-02-28 19:46:48 | 2021-02-28 19:49:04 | | 1.0 | 0.1 |

* Full path "/user_parameters/attached_worker_migration_bias". †Full path "/user_parameters/unattached_worker_migration_bias".

To allow jobs to run among a team concurrently, an online database must be used. For example, Google Sheets can be used as a database. Once the credentials for Sheets have been made, line 5 of Listing 2 needs changed to `db = dapt.db.Sheets(config=config)`, assuming the credentials are stored in the configuration file. Then multiple team members can execute the script locally, completing the set of jobs concurrently. DAPT will determine which parameters each person should run when the `next_parameter()` method is called. The `sheets_example.py` script [12] in the PhysiCell example repository demonstrates how Google Sheets can be used as the database. Google Sheets Example Section of the README [13] outlines required changes to paper_example.py to enable use of Sheets and how to setup Google Sheets.

## Future directions

In the next version of DAPT, we plan to implement logging using the Python `logging` library. Logging is useful for keeping track of errors and providing more detail for debugging. Additionally, we will allow notifications to be sent to users when certain events have occurred. For example, an email or Slack notification could be sent out when there are no more parameters to test. We would like to create a web interface to make managing parameter sets easier. Online spreadsheet programs like Google Sheets have user friendly interfaces, but these spreadsheets can be difficult to manage as the number of parameters

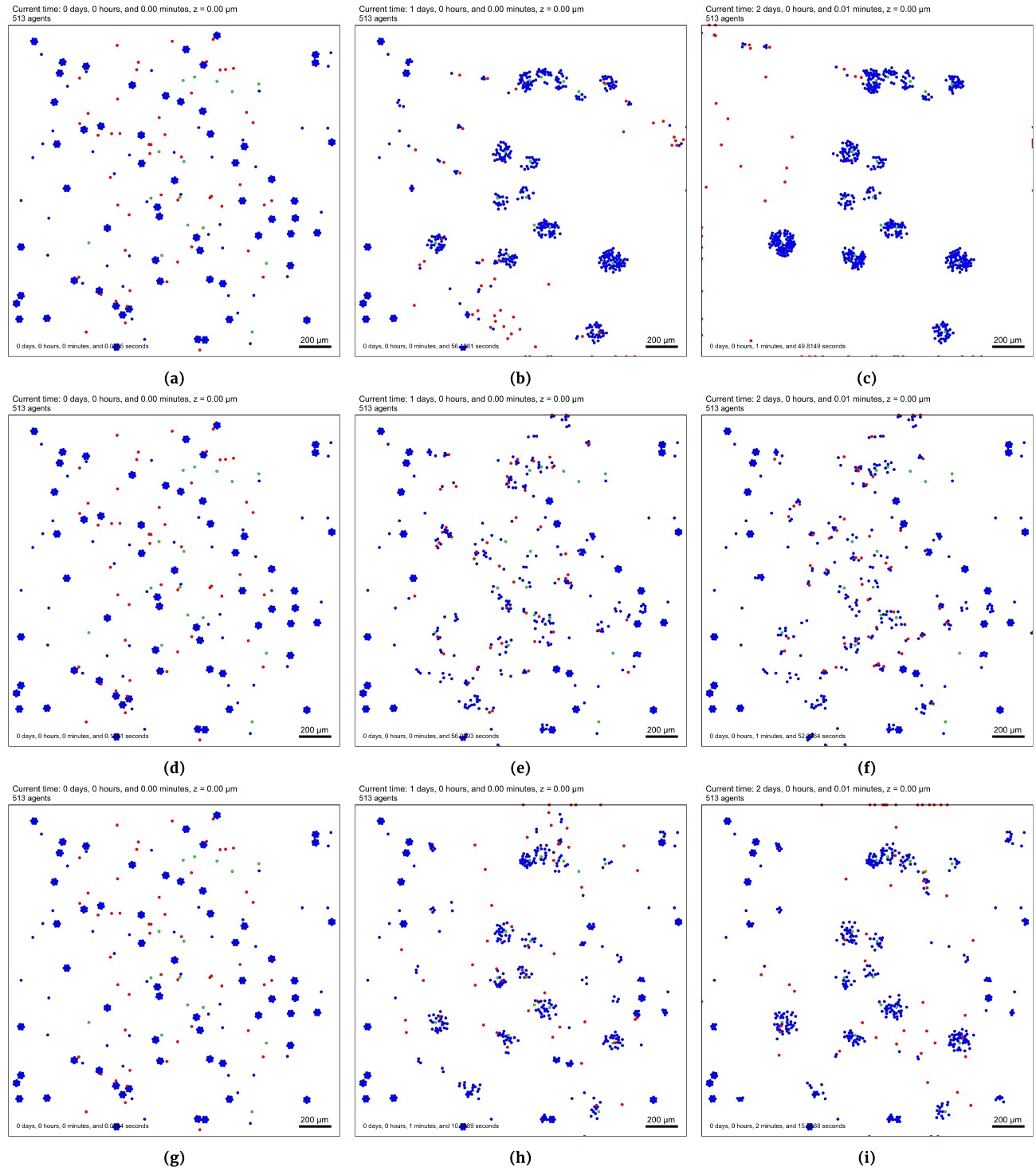

**Figure 2.** PhysiCell time series snapshots obtained by DAPT running the three parameter sets on the Biorobots sample project. The red (worker) cells drag the blue (cargo) cells towards a green (cancer) cell in an effort to treat the cancer. (a)–(c) Initial, middle (1 day of simulated time) and final (2 days of simulated time) image outputs of the *default* Biorobots simulation settings. (d)–(f) The same outputs as (a)–(c) but for the *attached* Biorobots simulation settings. (g)–(i) The same outputs as (a)–(c) but for the *unattached* Biorobots simulation settings.

grows. We also plan to integrate different APIs at a lower level to allow bots (e.g., Slack Bot) to generate notifications and control parameter testing. Furthermore, we plan to allow DAPT to be used in a tool via a command line interface (CLI). The Python scripting capability will not be removed, as having that level of control can be desirable. However, using DAPT directly in a CLI should increase efficiency in developing a testing pipeline.

## AVAILABILITY OF SOURCE CODE AND REQUIREMENTS

DAPT (RRID:SCR_021032) is primarily hosted on GitHub [14]. It is licensed under the BSD 3-clause license. All operating systems that support Python versions 3.6 through 3.9.1 (most recent version at time of publishing) can run DAPT. The best way to install DAPT is by using the Python Package Index (pip) version 20.2.4 or newer. To install DAPT run `pip install dapt` in the terminal (Linux/Mac OS) or command prompt (Windows). DAPT can also be installed from source. The documentation for DAPT is hosted on ReadTheDocs [15].

## DATA AVAILABILITY

Snapshots of our code and other supporting data are openly available in the GigaScience Repository, GigaDB [16].

## DECLARATIONS
## LIST OF ABBREVIATIONS

CLI: command-line interface; CSV: comma separated values; DAPT: Distributed Automated Parameters Tester; F@H: Folding @Home; HPC: high-performance computing; JSON: JavaScript Object Notation; ME: model exploration.

## ETHICAL APPROVAL

Not applicable.

## COMPETING INTERESTS

The authors declare that they have no competing interests.

## FUNDING

We thank the Jayne Koskinas Ted Giovanis Foundation for Health and Policy for generous support. This work was partially supported by the National Science Foundation NRT Grant 1735095.

## AUTHOR'S CONTRIBUTIONS

BSD conceived the main idea for the project and wrote the software. BSD and JPM tested the code extensively. JPM and PM provided mentorship during the project. BSD and JPM wrote the manuscript with help from PM.

## ACKNOWLEDGEMENTS

We would like to thank Daniel Murphy and Brandon Fischer for helping with the design and initial testing of DAPT. Thanks to Randy Heiland and the rest of the MathCancer lab for their help and feedback on the project. Their feedback was greatly appreciated during the development of this tool. FutureSystems, located at the Digital Science Center, Luddy School of Informatics, Computing, and Engineering, Indiana University, was an invaluable resource while developing DAPT.

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
