## [Reviewer Report]

Reviewer name and names of any other individual's who aided in reviewerMeghna VermaDo you understand and agree to our policy of having open and named reviews, and having your review included with the published manuscript. (If no, please inform the editor that you cannot review this manuscript.)YesIs the language of sufficient quality?YesPlease add additional comments on language quality to clarify if neededIs there a clear statement of need explaining what problems the software is designed to solve and who the target audience is? YesAdditional CommentsThe link for documentation (https://dapt.readthedocs.io/en/latest/dapt-api/param.html#fields) for the list of Database attributes shows an error message "(This page does not exist)"Is the source code available, and has an appropriate Open Source Initiative license <a href="https://opensource.org/licenses" target="_blank">(https://opensource.org/licenses)</a> been assigned to the code?YesAdditional CommentsAs Open Source Software are there guidelines on how to contribute, report issues or seek support on the code?YesAdditional CommentsIs the code executable?Unable to testAdditional Comments- The authors have nicely explained the paper_example.py in README and Nanohub link works (Introduction: a simulation of biorobots). 
- I was able to download and run PhysiCell on Windows using sample files (omp_test.cpp) listed in QuickStart.pdf here: https://github.com/MathCancer/PhysiCell (MinGW did not work for 64 bit, Cygwin worked). 

- However, when running 'make' from https://github.com/PhysiCell-Tools/DAPT-example, it gave an error: g++ -march=native -O3 -fomit-frame-pointer -mfpmath=both -fopenmp -m64 -std=c++11 -c ./BioFVM/BioFVM_vector.cpp
In file included from ./BioFVM/BioFVM_vector.h:55,
from ./BioFVM/BioFVM_vector.cpp:49:
/usr/lib/gcc/x86_64-pc-cygwin/10/include/c++/cmath:1065:11: error: ‘double_t’ has not been declared in ‘::’
1065 | using ::double_t;
| ^~~~~~~~
/usr/lib/gcc/x86_64-pc-cygwin/10/include/c++/cmath:1066:11: error: ‘float_t’ has not been declared in ‘::’
1066 | using ::float_t;
| ^~~~~~~
make: *** [Makefile:88: BioFVM_vector.o] Error 1
). Is installation/deployment sufficiently outlined in the paper and documentation, and does it proceed as outlined?YesAdditional CommentsIs the documentation provided clear and user friendly?YesAdditional CommentsIs there a clearly-stated list of dependencies, and is the core functionality of the software documented to a satisfactory level?YesAdditional CommentsMinGW-x64 does not work on Windows, Cygwin seemed to work okay. Have any claims of performance been sufficiently tested and compared to other commonly-used packages? Not applicableAdditional CommentsThe package has been tested for PhysiCellAre there (ideally real world) examples demonstrating use of the software? YesAdditional CommentsIs automated testing used or are there manual steps described so that the functionality of the software can be verified?YesAdditional CommentsAny Additional Overall Comments to the Author- In the introduction section, it would be helpful to spell out EMEWS and OpenMOLE. The authors can also briefly describe EMEWS and OpenMOLE. 
In- Fourth paragraph, a review related to the client and database server might help.
- The link for documentation for the list of Database attributes shows an error message '(This page does not exist)'
- Suggestion: 
o In the Implementation section, 2nd paragraph regarding Database description, citation of Table 2 can help with visualization. 
o In the example section, “The folder structure of this project…The first two lines of code [mention Listing 2]. 
- Last paragraph of the Implementation section, “Once the credentials…” is it line 6 instead of line 4?
- Future directions, can authors comment on the inclusion of replicates for each set of parameters?RecommendationMinor Revisions

---

## [Reviewer Report]

Reviewer name and names of any other individual's who aided in reviewerBoris AguilarDo you understand and agree to our policy of having open and named reviews, and having your review included with the published manuscript. (If no, please inform the editor that you cannot review this manuscript.)YesIs the language of sufficient quality?YesPlease add additional comments on language quality to clarify if neededIs there a clear statement of need explaining what problems the software is designed to solve and who the target audience is? YesAdditional CommentsIs the source code available, and has an appropriate Open Source Initiative license <a href="https://opensource.org/licenses" target="_blank">(https://opensource.org/licenses)</a> been assigned to the code?YesAdditional CommentsAs Open Source Software are there guidelines on how to contribute, report issues or seek support on the code?YesAdditional CommentsIs the code executable?YesAdditional CommentsIs installation/deployment sufficiently outlined in the paper and documentation, and does it proceed as outlined?YesAdditional CommentsIs the documentation provided clear and user friendly?YesAdditional CommentsIs there a clearly-stated list of dependencies, and is the core functionality of the software documented to a satisfactory level?YesAdditional CommentsHave any claims of performance been sufficiently tested and compared to other commonly-used packages? NoAdditional CommentsThe authors listed a couple of commonly-used systems for HPC based parameter exploration. Since this was intended for user not necessarily familiar with HPC, a comparison was not performed. Are there (ideally real world) examples demonstrating use of the software? YesAdditional CommentsThere are multiple basic examples; but I think that should mentioned in the main text. Is automated testing used or are there manual steps described so that the functionality of the software can be verified?YesAdditional CommentsAny Additional Overall Comments to the AuthorRecommendationMinor Revisions